# Lactate Can Modulate the Antineoplastic Effects of Doxorubicin and Relieve the Drug’s Oxidative Damage on Cardiomyocytes

**DOI:** 10.3390/cancers15143728

**Published:** 2023-07-22

**Authors:** Valentina Rossi, Marzia Govoni, Giuseppina Di Stefano

**Affiliations:** Department of Medical and Surgical Sciences (DIMEC), Section of General Pathology, University of Bologna, 40126 Bologna, Italy; valentina.rossi78@unibo.it (V.R.); marzia.govoni@unibo.it (M.G.)

**Keywords:** lactate, cancer cell metabolism, doxorubicin, oxidative stress

## Abstract

**Simple Summary:**

Cancer cells are characterized by massive glucose consumption, leading to increased lactate generation. Once considered a waste product, this metabolite was recently shown to take part in the regulation of gene expression. The regulatory properties of lactate were also found to play a role in fostering drug resistance. In this paper, we examined whether the exposure of cancer cells to increased lactate levels can affect the anticancer efficacy of doxorubicin. Doxorubicin is a widely used drug, whose clinical application is often hampered by severe cardiotoxicity. We found that lactate can offer protection against oxidative damage caused by the drug. Interestingly, oxidative damage is reputed secondary to the antineoplastic action of the drug but plays an important role in mediating its cardiotoxic effects.

**Abstract:**

Background: Doxorubicin (DOXO) is currently administered as the first-choice therapy for a variety of malignancies. Cancer cells exhibit enhanced glycolysis and lactate production. This metabolite affects gene expression and can play a role in chemoresistance. Aim of this study: We investigated whether the enhanced lactate levels that characterize neoplastic tissues can modify the response of cancer cells to DOXO. Methods: After exposing cancer cells to increased lactate levels, we examined whether this metabolite could interfere with the principal mechanisms responsible for the DOXO antineoplastic effect. Results: Increased lactate levels did not affect DOXO-induced topoisomerase poisoning but offered protection against the oxidative damage caused by the drug. This protection was related to changes in gene expression caused by the combined action of DOXO and lactate. Oxidative damage significantly contributed to the heavy cardiotoxicity following DOXO treatment. In cultured cardiomyocytes, we confirmed that DOXO-induced DNA damage and oxidative stress can be significantly mitigated by exposing the cells to increased lactate levels. Conclusions: In addition to contributing to elucidating the effects of the combined action of DOXO and lactate, our results suggest a possible method to reduce the heavy drug cardiotoxicity, a major side effect leading to therapy discontinuation.

## 1. Introduction

To improve neoplastic disease treatment, recent efforts have mainly focused on the development of new-generation drugs, targeting critical pathways involved in cancer cell proliferation and survival. However, despite the unquestionable progress achieved with these studies, the currently applied anticancer therapeutic regimens very often continue to maintain the use of old, conventional chemotherapeutic agents. In this context, the enduring use of doxorubicin (DOXO) in the fight against disparate cancer diseases is exemplificative [1,2]. 

DOXO belongs to the family of anthracyclines and is probably one of the most effective antineoplastic drugs ever developed, even though its potential side effects (heart muscle damage) can limit the success of treatment [3]. DOXO and its variants are currently administered as the first-choice therapy for a variety of solid and hematologic malignancies and have also been evaluated in combination treatments [1,3]. 

Concurrently with this extensive and long-lasting clinical use, the molecular targets of DOXO action have been actively investigated and a number of potential antineoplastic mechanisms have been proposed for this drug. Some of these postulated mechanisms are probably not relevant for the antineoplastic effects observed in clinics since they were documented at DOXO concentrations far above those reached in the blood circulation of treated patients. Others were observed at clinically relevant drug doses [4,5,6]. Among these, the best characterized are topoisomerase-2 (TOP2) poisoning, free radical generation, ceramide overproduction and histone eviction. In addition to contributing to DOXO antineoplastic action [7], free radical generation is also considered to be involved in heart muscle damage [8], often hindering an adequately lasting treatment. Histone eviction from nucleosomes results from the well-documented capacity of DOXO to intercalate between DNA bases [5,6]; DOXO intercalation causes torsional stress in the DNA helix, leading to H2A/H2B dissociation. This event could be involved in the antineoplastic effects and could also explain the extensive alteration in transcriptome observed in cells exposed to DOXO or to its variants [9].

Changes in cell transcriptome can also be linked to cellular metabolic asset since intermediates originating from energy metabolism reactions have been found to alter histone acetylation and chromatin structure [10]. One of the most studied metabolites showing the potential of altering gene expression is lactate, the end-product of glycolysis. In addition to inhibiting histone deacetylases (HDAC) enzymes [11,12], lactate was shown to promote gene expression through histone lactylation [13], a modification that facilitates the access of transcription complexes on DNA. In this way, lactate can function as a sensor able to transduce metabolic changes into stable gene expression patterns.

Experiments performed in our laboratory showed that lactate has the potential to reduce the antineoplastic efficacy of cisplatin [14], an effect linked to the increased expression of repair pathways’ genes usually involved in the management of DNA alterations caused by this drug. Furthermore, we observed that lactate can play an active role in promoting tamoxifen resistance of breast cancer cells [15].

The experiments reported in this paper investigated whether this metabolite could interfere with the principal mechanisms responsible for the DOXO antineoplastic effect, considering the transcriptional changes potentially caused by the joint action of both compounds.

The experimental outline is summarized in the following Figure 1:

## 2. Materials and Methods

### 2.1. Cell Cultures and Treatments

All the materials used for cell culture and all the reagents were obtained from Merck (Darmstadt, Germany) unless otherwise specified. MCF7, MDA-MB-231, CaCo2 and H9c2 cells (ATCC, Manassas, VA, USA) were grown in low-glucose (5 mM) Dulbecco’s minimal essential medium (DMEM), supplemented with 100 U/mL penicillin/streptomycin, 2 mM glutamine and 10% fetal bovine serum (FBS). Lactate (L-isomer) was dissolved in the culture medium at a 20 mM concentration; before DOXO treatment, MCF7 and H9c2 cell cultures were exposed to 20 mM lactate for 72 h. DOXO (Selleckchem, Huston, TX, USA) was dissolved in dimethyl-sulfoxide (DMSO). In DOXO including experiments, the culture medium was supplemented with 0.6% DMSO. Differentiation of H9c2 cardiomyocytes was induced when cells reached 80% confluence; to this aim, cells were maintained for 10–11 days in DMEM with 1% FBS and supplemented daily with 10 nM retinoic acid (Enzo Life Sciences, Farmingdale, NY, USA).

### 2.2. Assay of Lactate Levels

Cells (5 × 10^5^/well) were plated in triplicate in 6-well plates and left to adhere overnight. The medium was then replaced with Krebs–Ringer buffer and the released lactate was measured after 2–4 h incubation at 37 °C using the method described in [16]. Briefly, at the end of incubation, 200–400 μL of Krebs–Ringer buffer was withdrawn, brought to 1 mL with H_2_O and mixed with 200 μL of 20% CuSO_4_; 200 mg di Ca(OH)_2_ were then added, and the tubes were maintained for 30 min at room temperature. After a 5 min centrifugation at 400 rpm, 200 μL of the supernatant was placed in conical glass tubes containing 1.25 mL of 96% H_2_SO_4_, vortexed and boiled for 5 min at 100 °C. The samples were then placed on ice and added with 6.4 μL of 4% CuSO_4_ and 5 μL of phenyl–phenol. After 30 min incubation at 37 °C and 1.5 min of boiling, they were left to cool at room temperature. The absorbance was read at λ_565_, and the amount of lactic acid in the samples was calculated with the aid of a calibration curve prepared with known amounts of sodium lactate.

### 2.3. Immunoblotting

Immunoblotting was used (a) to evidence DOXO-induced DNA damage, which was assessed with histone 2AX phosphorylation (γ-H2AX, an established marker of DNA damage [17]); (b) to evaluate, in treated cells, the levels of lactate dehydrogenase A (LDHA), glucose-regulated protein 94 (GRP94, also known as heat shock protein 90B), superoxide dismutase 1 and 2 (SOD1, SOD2) and single-strand selective monofunctional uracil DNA glycosylase (SMUG1); (c) to assess the level of acetylated histone 3 in lactate-exposed MCF7 cells.

For evidencing γ-H2AX, LDHA and GRP94 proteins, the cell cultures were exposed to 1 μM DOXO from 0 to 24 h. SOD1, SOD2 and SMUG1 proteins were evaluated in control and lactate-exposed MCF7 cells treated with 1 μM DOXO for 4 h, followed by a 16 h recovery time.

In all immunoblotting experiments, at the indicated times after treatment, cell cultures (T-25 flasks, at 80% confluence) were harvested and lysed in 50 μL RIPA buffer containing protease and phosphatase inhibitors. Then, 60 μg of protein (measured according to Bradford) was loaded onto precast 4–12% polyacrylamide gels for electrophoresis and run at 170 V. The separated proteins were blotted on low-fluorescent PVDF membranes (Cytiva Life Sciences, Milano, Italy) using a standard apparatus for wet transfer with an electrical field of 60 mA for 16 h. The blotted membranes were blocked with 5% BSA in TBS-Tween and probed with the primary antibody. Actin was used as a loading control in all experiments, with the exception of those performed in MCF7 cultures and requiring long exposure times (≥6 h) to DOXO. In agreement with published data [18], we observed that long DOXO exposures changed the actin level in MCF7 cells; in these cases, constitutive heat shock 70 (HSC70) was the selected internal control protein. For each immunoblotting experiment, the used internal control protein can be verified from the Figures (Figures 1, 3, 4, 6 and 7) by referring to the indicated molecular weights (38–49 kDa and 62–98 kDa for actin and HSC70, respectively).

The used antibodies were: rabbit anti-human γ-H2AX (phospho-S139) (Abcam, Cambridge, UK); rabbit anti-rat γ-H2AX (phospho-S139) (Cell Signaling, Leiden, NE); rabbit anti-LDH-A (Cell Signaling); rabbit anti-GRP94 (Cell Signaling); rabbit anti-histone-3 (Cell Signaling); rabbit anti-panacetyl-histone-3 (Active Motif, Brussels, BE); rabbit anti-human and rat SOD1 (Cohesion Bioscience, London, UK); rabbit anti-human and rat SOD2 (Cohesion Bioscience); rabbit anti-SMUG1 (Cohesion Bioscience); rabbit anti-Actin (Merck); and rabbit anti-HSC70 (Enzo Life Sciences). Binding was revealed using a Cy5-labelled secondary antibody (goat anti-rabbit-IgG, Cytiva Life Sciences). The fluorescence of the blots was assessed with the Pharos FX Scanner (Bio-Rad, Hercules, CA, USA) at a resolution of 100 μm. The intensity of the bands was evaluated using the ImageJ 1.53a software methods.

### 2.4. Evaluation of Oxidative Stress

Oxidative stress was evaluated using a 2′,7′-dichlorofluorescin diacetate (DCF-DA) assay. After cell entry, in the presence of reactive oxygen species (ROS), this probe is oxidized to highly fluorescent dichloro-fluorescein (DCF) [19].

Control and lactate-exposed MCF7 cells were grown on coverslips in 6-well plates and treated for 2 h with 1 μM DOXO. After washing with PBS two times, the cells were incubated at room temperature in the dark for 25 min with 10 μM DCF-DA dissolved in PBS. Cultures were then quickly washed with PBS and mounted with a solution of Hoechst/DABCO (1:200). The samples were observed using a Nikon epifluorescence microscope equipped with filters for Hoechst and FITC. Cells showing a bright and intense fluorescence were counted as positive, whereas cells having no or low fluorescence were counted as negative. For each sample, 500–700 cells were counted by two independent observers. A similar experiment was performed using proliferating and differentiated H9c2 (D-H9c2) cells. In this case, the control and lactate-exposed cells were treated with 1 μM DOXO for 16 h.

In a further experiment, oxidative stress in DOXO-treated MCF7 cells was evaluated by measuring 8-hydroxy-2-deoxy guanosine (8-OHdG), a marker of DNA oxidative damage, generated by reactive oxygen and nitrogen species [20]. A commercially available competitive ELISA test was used (Abcam) following the manufacturer’s protocol. Control and lactate-exposed MCF7 cells were treated with 1 μM DOXO for 2 h. Genomic DNA was isolated using the phenol/chloroform/isoamyl alcohol extraction procedure [21]. DNA was digested using Nuclease P1 (New England BioLabs, Ipswich, MA, USA) in 50 mM sodium acetate pH 5.5; the pH was then adjusted to 7.5–8.5 using 1 M TRIS. For each sample, 100 μg of DNA was incubated at 37 °C for 30 min with 1 unit of alkaline phosphatase. Samples were then boiled for 10 min and placed on ice until used. For the ELISA procedure, they were diluted to a final DNA concentration of 500 ng/mL, and 25 ng samples were used for the assay. At the end of the procedure, absorbance was measured using a Victor plate reader (PerkinElmer, Waltham, MA, USA) with a wavelength of 450 nm.

### 2.5. Real-Time PCR

This experiment was performed in control and lactate-exposed MCF7, MDA-MB-231 and CaCo2 cells to evaluate the expression of inducible heat shock protein 70 (HSP70), GRP94 and TNF receptor-associated protein 1 (TRAP1, the mitochondrial form of HSP90). Exponentially growing cells from T25 flasks were used. In a further, different experiment, control and lactate-exposed MCF7 cells were treated with 1 μM of DOXO for 4 h followed by a 16 h recovery. RNA was extracted using an RNA isolation kit (Merck) and was quantified spectrophotometrically (ONDA Nano Genius Photometer). Retro-transcription to cDNA was performed using a Revert Aid First Strand cDNA Synthesis Kit (ThermoFisher, Waltham, MA, USA) in different steps: 5 min denaturation at 65 °C, 5 min annealing at 25 °C, 1 h retro-transcription at 42 °C and 5 min at 70 °C. A real-time PCR (RT-PCR) analysis was performed using 20 ng cDNA, SsoAdvanced Universal SYBR Green Supermix (Bio-Rad, Hercules, CA, USA) and different primer mixtures. All the primers used for the PCR experiments were predesigned and obtained from Merck. For all genes, the annealing temperature of the primers was 60 °C, and the thermal cycler (CFX96 TM Real-Time System, Bio-Rad) was programmed as follows: 30 s at 95 °C, 40 cycles of 15 s at 95 °C and 30 s at 60 °C. The data from RT-PCR experiments were analyzed by applying the 2^−ΔΔCT^ method [22].

### 2.6. Quantification of TOP2A:DNA Covalent Complexes

To assess a possible increase in TOP2A:DNA complexes after DOXO treatment, the in vivo complex of the enzyme assay described in [23] was followed, with some modifications.

Briefly, control and lactate-exposed MCF7 cells (T-25 flasks, at 80% confluence) were treated with 1 μM DOXO for 24 h. After treatment, the medium was removed and 3 mL of 1% Sarkosyl in 1X TE pH 7.5 (10 mM Tris HCl pH 7.5, 1 mM EDTA) was added to the flasks. The addition of the anionic detergent caused cell lysis and stabilization of the DNA–protein complexes, which became measurable. Cell lysates were sheared with a syringe equipped with a 25G gauge needle, and the final volumes were then increased to 10 mL with 1% Sarkosyl solution. Sheared lysates were gently stratified onto 3 mL of 9 M CsCl, dissolved in H_2_O in 13.5 mL Quick-Seal centrifuge tubes (Beckman Coulter, Pasadena, CA, USA) and centrifuged (121,900× *g* for 21 h at 25 °C). The pelleted material containing DNA and bound proteins was washed once with 500 μL of 70% ethanol, air-dried to remove ethanol, dissolved again in 200 μL 1X TE buffer pH 7.5, left overnight at 4 °C and then incubated in a water bath at 65 °C for 5 min to complete resuspension.

The DNA concentration of the samples was measured; scalar amounts of DNA (0–5 μg) were diluted to a final volume of 200 μL with 1X TE buffer and applied on a PVDF membrane (Cytiva Life Science) using a dot-blot apparatus (Bio-Rad). The blotted membrane was blocked with 5% BSA (in TBS-TWEEN) for 1 h and probed with the primary antibody (rabbit-anti-topoisomerase II alpha, Abcam). Binding was revealed using a Cy5-labelled secondary antibody (goat anti-rabbit-IgG, Cytiva Life Sciences). The fluorescence of the blots was assessed with the Pharos FX Scanner (Bio-Rad) at a resolution of 100 μm. The results were evaluated with densitometry using the Protein Array Analyzer in ImageJ software with the aid of the Gilles Carpentier’s Dot-Blot-Analyzer macro (2008). This macro is available at http://rsb.info.nih.gov/ij/macros/toolsets/Dot%20Blot%20Analyzer.txt, accessed on 1 January 2020.

### 2.7. Cell Proliferation

These experiments were performed in control and lactate-exposed MCF7 cultures and in an MCF7 subclone grown in the presence of 20 mM lactate for at least 6 months (LAC-MCF7) [12]. Cells (20 × 10^4^ cells/well) were plated in clear bottom 96-well white plates, left to adhere overnight and then exposed to scalar doses of DOXO (0–8 μM) for 24–48 h. At the end of treatment, cell proliferation was assessed with the detection of ATP levels using the CellTiter-Glo Assay (Promega, Madison, WI, USA). A Fluoroskan Ascent FL reader was used to evaluate the plates’ luminescence.

### 2.8. Statistical Analysis

All data were analyzed using the GraphPad Prism 5 software. All results were obtained from at least two independent experiments performed with duplicate or triplicate samples. The results are expressed as mean values ± SE and were calculated using all the data obtained from the independent experiments; the significance level was set at *p*  <  0.05.

## 3. Results

### 3.1. Enhanced Lactate Levels Modified the DNA Damaging Effects Caused by DOXO

The primary model used for our study was the MCF7 breast cancer cell line. Chemotherapeutic treatments based on DOXO are a frequent therapeutic option for patients diagnosed with breast cancer [24], and MCF7 cultures are among the most studied models for this neoplasm. These cells are characterized by moderate-level glycolysis, which allows to easily evidence lactate-induced cell changes following exogenous administration of increased amounts of this metabolite [15]. To highlight a possible role of lactate in modifying DOXO antineoplastic action, MCF7 cultures were routinely grown with 5 mM glucose and exposed to 20 mM lactate, added to the medium 72 h before experiments. This lactate concentration was previously shown to facilitate the onset of tamoxifen resistance [15] and fits well with the level of metabolite usually assessed in the microenvironment of neoplastic tissues [25]. The data obtained from MCF7 cultures exposed to lactate and treated with DOXO were compared with results acquired for the MDA-MB-231 and CaCo2 cultures. MDA-MB-231 cells are a widely used model of drug-resistant breast cancer [26]. The CaCo2 culture was selected as representative of a neoplastic disease from a different tissue, for which DOXO-based treatment is a frequent therapeutic option [27]. As shown in Figure 1A, both these cultures were characterized by significantly increased levels of glycolysis and lactate production compared to MCF7 cells (*p* < 0.01 at both 2 and 4 h).

In the first experiment, the three cell cultures, together with lactate-exposed MCF7 cells, were treated with 1 μM DOXO, and the drug-induced DNA damage was assessed at different times using an immunoblotting evaluation of γ-H2AX [17]. The results are shown in Figure 1B,C. Interestingly, all the cultures exposed to higher lactate levels (derived from either constitutively increased glycolysis rate or exogenous administration) appeared to respond similarly to DOXO treatment. In contrast, they showed an appreciably reduced DNA damage signal compared to MCF7 cultures at 2 and 6 h. In lactate-exposed MCF7 cells, the increase in γ-H2AX band intensities measured at these time intervals after DOXO treatment was 2.4- and 1.8-fold lower than that measured in DOXO-treated parental MCF7 cells (*p* < 0.01 and < 0.05 at 2 and 6 h, respectively). Conversely, the fold increase in the DNA damage signal assessed at 16 and 24 h appeared to be significantly higher in the lactate-exposed MCF7 culture (2.2-fold at 24 h with *p* < 0.05) (Figure 1C).

The immunoblotting experiment shown in Figure 1B,C also investigated the changes in LDHA level during the phase of DOXO treatment. As well known, DNA damage triggers glycolysis [28] since increased ATP levels are required for the repair reactions. Interestingly, parental MCF7 and MDA-MB-231 cells showed progressively enhanced LDHA levels, in relation to the timing of DOXO exposure: in the case of MCF7 cells, a ≈2.3-fold increase was measured at both 16 and 24 h (*p* < 0.05). This increase appeared to be linked to lactate levels since it was undetected in lactate-exposed MCF7 cells and in CaCo2, the cell line showing the highest rate of lactate production (Figure 1A). This result could be interpreted as a feedback regulation mechanism induced by elevated lactate levels in the medium. Increased LDHA is expected to lead to higher lactate production, which could potentially affect cell response to DOXO. The viability experiments described below (see Section 3.5) showed no difference between lactate-exposed MCF7 cells and their parental culture treated with DOXO for 24–48 h. This result suggests that the evidenced glycolytic flare (Figure 1B,C) could play a role in equalizing the drug response of the two cell cultures in the long term.

### 3.2. Enhanced Lactate Levels Reduced Free Radical Generation Caused by DOXO

According to the data shown in Figure 1B, lactate exposure appeared to reduce the early DNA damage (2–6 h) caused by DOXO. For their rapid onset, the DNA damage signatures detected in parental MCF7 cells at this time could be attributed to drug-induced oxidative stress, a mechanism mainly accounting for the detrimental side effects of DOXO [7]. This hypothesis is also in agreement with the notion that cancer cells with activated glycolysis (such as MDA-MB-231 and CaCo2 cells) usually show protection against oxidative stress [29]. This was verified with the experiments shown in Figure 2.

Parental and lactate-exposed MCF7 cells were grown on glass slides and treated with 1 μM DOXO for 2 h, and ROS generation was visualized after applying a DCFH-DA probe. The labeled cells were identified on pictures taken using a microscope after merging the fluorescent DCF signal with Hoechst nuclear staining (Figure 2A). In agreement with the data obtained with the immunoblotting evaluation of γ-H2AX, the bar graph in Figure 2B shows that lactate exposure resulted in a significantly dampened ROS generation in DOXO-treated MCF7 cells: the percentage of labeled cells in control cultures was reduced to about one-third by lactate (42 % vs. 14%, *p* < 0.01).

Additional evidence for the reduced ROS generation in lactate-exposed cells was achieved with the experiment shown in Figure 2C. As well documented, ROS can modify guanine bases in DNA, giving rise to 8-OHdG. This modified guanine is considered an oxidative stress biomarker [20]. Moreover, 8-OHdG easily forms base pairs with thymidine, and this property could contribute to the mutagenic activity of DOXO [20]. The bar graph in Figure 2C shows that DOXO treatment caused a ≈38% increased level of 8-OHdG in parental MCF7 cells, while only an ≈11% increase was detected following lactate exposure. The subsequent step in our experiment was aimed at identifying possible molecular mechanisms underlying the oxidative damage protection effect observed in lactate-exposed cells.

### 3.3. Involvement of the Stress Response

Recently published studies showed that in tumor cells, high lactate production is often associated with stress protein overexpression and radio-resistance [30,31]; on the contrary, depletion of lactate by LDH inhibition or knockout was found to impair stress response and promote cell sensitivity to ionizing radiation [32]. The application of ionizing radiation in tissues is typically followed by the induction of oxidative stress and ROS generation [33]; therefore, we hypothesized that the results observed in lactate-exposed MCF7 cells (Figure 1 and Figure 2) would be mediated by a lactate-induced stress response. For this reason, we verified whether the increased lactate levels in MCF7 cells could promote the expression of three stress proteins, for which a role in the protection from DNA damage induced by oxidants is well documented: inducible HSP70 and two members of the HSP90 family, GRP94 and TRAP1 [34,35,36]. Furthermore, in previous studies, HSP70 and GRP94 were found to be linked with glycolytic metabolism, and their increased levels were associated with poor survival and worse prognosis in breast cancer [37,38].

Figure 3A shows the mRNA level of the selected stress proteins as assessed using RT-PCR for all cultures used in the experiments shown in Figure 1 and Figure 2. No difference was observed between the used cell lines concerning HSP70 and TRAP1. In the cultures characterized by high lactate levels, the only observed upregulation concerned GRP94 mRNA, for which a ≈3-fold increased expression was measured in MDA-MB-231 and CaCo2 cells compared to the MCF7 culture. A modest but statistically significant increase was also observed in lactate-exposed MCF7 cells compared to their parental culture. However, when GRP94 levels were evaluated using immunoblotting (Figure 3B), the differences between the cell lines appeared to be clearly decreased, and no statistically significant finding was observed in lactate-exposed MCF7 compared to their parental culture. Since stress protein mRNAs are expected to be translated mainly as a consequence of harmful stimuli, the immunoblotting evaluation of GRP94 was repeated in DOXO-treated cells using the same conditions applied in the experiments shown in Figure 1B. The results are shown in Figure 3C,D. Again, no evident difference was observed between lactate-exposed and parental MCF7 cells. This was also shown by the levels of band intensities measured after 24 h of DOXO exposure, which, for all four cell cultures, were related to the GRP94 band intensity measured at T = 0 (Figure 3D).

Taken together, the experiments shown in Figure 3B,D suggest that, compared to MCF7 cells, the highly glycolyzing cultures express a slight but statistically significant level of the GRP94 protein. However, the data obtained from lactate-exposed MCF7 cultures suggest that this finding cannot be simply explained by the increased lactate level.

From all the experiments shown in Figure 3, we concluded that the three considered stress proteins are not involved in the oxidative damage protection observed in DOXO-treated and lactate-exposed MCF7 cells (Figure 2) or in the different time courses of DOXO-induced DNA damage shared by all the cultures characterized by increased lactate levels (Figure 1B,C).

### 3.4. Changes in Gene Expression following Lactate and DOXO Administration in MCF7 Cells

Both lactate and DOXO have the potential to modify gene expression [9,10,11,12,13]. Lactate was originally shown to increase histone acetylation [11], and, as shown in Figure 4A, this effect was also evidenced in lactate-exposed MCF7 cells. On the other hand, for its capacity to intercalate between DNA bases, DOXO was shown to cause the dissociation of histones/DNA complexes [6]. For these properties, the combined action of these two molecules can be expected to have a significant impact on cell transcriptome. As a following step, to explain the results shown in Figure 1 and Figure 2, we next verified whether lactate and DOXO could modify the expression of a number of genes involved in the oxidative stress response when given to MCF7 cells independently or in combination. Figure 4B,C shows the results of RT-PCR experiments aimed at evaluating the expression levels for a panel of genes involved in both oxidative damage repair (B) and protection (C). For these experiments, parental and lactate-exposed MCF7 cells were treated with 1 μM DOXO for 4 h; mRNA extraction for RT-PCR was performed after an additional 16 h rescue time. As expected, both DOXO and lactate appeared to affect gene expression levels, sometimes in opposite ways. When evaluating these results, we only considered genes showing upregulation caused by both lactate and DOXO and for which the combination of the two compounds resulted in a statistically significant increased gene expression. This criterion was used to obtain better evidence for the combined effect of the two compounds and was met by SMUG1, SOD1 and SOD2. The level of these proteins was further analyzed in treated cells using immunoblotting; the results are shown in Figure 4D,E. When given to MCF7 cells independently, lactate and DOXO did not affect SOD1 and SOD2 protein levels, which, however, appeared to be significantly increased in cells receiving the combined treatment (≈1.5 fold compared to control MCF7 cultures). After the immunoblotting evaluation, SMUG1 appeared to be split into two bands, compatible with two of the five isoforms described for this protein [39]. The higher molecular weight band is compatible with isoform 1 (NCBI Reference Sequence: NP_055126.1), whereas the lower is compatible with isoform 3 (NCBI Reference Sequence: NP_001338187.1). Isoform 3 originates from an alternative mRNA splicing and is missing part of the terminal sequence of SMUG1. Interestingly, the missing region contains amino acids found to participate in the catalytical reaction of SMUG1. For this reason, this protein isoform is expected to have compromised catalytic activity. When the band intensities of the two isoforms were analyzed together, no significant difference was observed between untreated and lactate/DOXO exposed cultures. After evaluating the relative abundance of the two SMUG1 isoforms, for the cells exposed to DOXO single treatment, we found a marked increase in the isoform 1 band, which reached 42% of the total protein signal, a value ≈7-fold higher than that observed in untreated cultures. Lactate was found to counteract the effect of DOXO on SMUG1 isoforms, and in cells exposed to the combined treatment, the more functional isoform of SMUG1 was ≈3-fold lower than measured in DOXO-treated cultures.

This observation suggests that despite the evidently increased SMUG1 mRNA expression found in cells exposed to the lactate/DOXO treatment, this protein should not be relevant to the oxidative damage protection observed in the cultures exposed to the combined treatment. In conclusion, and on the basis of our results, the observed protective effect can be only ascribed to the increased levels of SOD1 and SOD2. These findings provide a mechanistic explanation for the results observed in the experiments shown in Figure 1B,C and Figure 2.

Finally, the bar graph in Figure 4F shows a RT-PCR experiment aimed at evaluating the combined effect of DOXO and lactate on UDP-glucose ceramide glucosyltransferase (UGCG). UGCG (also cited as glucosylceramide synthase, GCS) is a rate-limiting enzyme in the synthesis of glycosylated sphingolipids [40], and its increased expression correlates with resistance to DOXO and other chemotherapeutic agents [41]. Figure 4F shows that DOXO treatment caused a ≈75% increased expression of UGCG; lactate did not affect the expression of this gene either when given as a single treatment or in combination with DOXO. From this experiment, we concluded that lactate should not impact the UGCG-based mechanism underlying DOXO resistance.

### 3.5. DOXO-Induced TOP2 Poisoning in Lactate-Exposed MCF7 Cells

TOP2 poisoning inhibition is considered the main antineoplastic mechanism of DOXO. TOP2 corrects DNA tangles that occur during replication and transcription by cleaving and resealing the filaments [42]. DOXO intercalates in DNA strands and prevents the resealing of cleaved sites, thus causing DNA double-strand breaks [43]. Specifically, TOP2A is the enzyme isoform found to be essential for cell proliferation, while TOP2B is mainly required for nervous tissue development. In DOXO resistance, an involvement of TOP1 was also described [44], and this form is also engaged in removing DNA supercoils during transcription and replication [45].

To assess whether lactate and DOXO can affect the expression of topoisomerase genes, an RT-PCR assay was performed using the same treatment schedule used for the experiments shown in Figure 4. The results (Figure 5A) showed a significantly increased expression of all three cited topoisomerase forms in DOXO-treated cells; no contribution related to lactate emerged in this experiment. To confirm that lactate does not affect the primary antineoplastic mechanism of DOXO, a dot-blot assay was performed to evaluate DNA-TOP2A complexes in control and lactate-exposed MCF7 cultures treated for 24 h with 1 μM DOXO. The results are shown in Figure 5B,C. For this experiment, samples containing scalar doses of DNA were used. The dot fluorescence was analyzed with ImageJ software using the built-in dot-blot-analyzer macro (see Section 2), which allowed us to obtain the heatmap shown in Figure 5C. For each DNA sample, the signal given by the DNA-trapped TOP2A in DOXO-treated cells was normalized on the respective untreated control and plotted on a graph. As shown in the graph in Figure 5C, the values for the fluorescence intensity ratios measured in control and lactate-exposed MCF7 cells did not show significant differences, suggesting that the level of TOP2A inhibition caused by DOXO in lactate-exposed cultures is superimposable to that observed in control MCF7 cells.

Finally, we performed cell viability experiments to evaluate the antiproliferative effect of DOXO in control and lactate-exposed MCF7 cells. Again, no difference was observed between the two cultures at either 24 or 48 h (Figure 5D). Taken together, the described experiments suggest that lactate should not impair the primary antineoplastic mechanism of DOXO, at least when lactate is administered using a short-term exposure before DOXO treatment. Interestingly, for the experiments shown in Figure 5D, we also used an MCF-7 sub-culture that was adapted to grow in the presence of 20 mM lactate for about 6 months (LAC-MCF7). In previous experiments [15], we observed that these cells showed evidence of an activated EGFR signaling pathway, a condition usually correlating with poor drug response [46]. As shown in the Figure, LAC-MCF7 cells showed a significantly reduced response to 1 μM DOXO. We hypothesize that this finding could reflect a phenomenon of neoplastic progression triggered by a continuative exposure to lactate [47], which would not be expected in the case of sporadic administrations of this metabolite.

### 3.6. Experiments on H9c2 Cardiomyocytes

While the contribution of DOXO-induced oxidative damage is considered secondary to its antineoplastic effect, there is consolidated evidence that it plays a major role in the cardiac toxicity caused by the drug [5,48]. Following the results obtained for lactate-exposed MCF7 cells, we wondered whether a similar protective effect against DOXO-induced oxidative damage could be exerted by lactate in cardiomyocytes. For this experiment, we used rat H9c2 cells, a validated model for studying DOXO cardiotoxicity and developing potential cardioprotective strategies [49]. The combined effects of DOXO and lactate on oxidative damage were studied in both proliferating and differentiated H9c2 cells (Figure 6 and Figure 7, respectively). For these experiments, H9c2 cells were exposed to 20 mM lactate for 72 h, following the same schedule used for lactate-exposed MCF7 cells. Figure 6A shows the results from the immunoblotting evaluation of γ-H2AX performed on proliferating H9c2 cells treated with 1 μM DOXO.

In these cells, evidence of DNA damage was observed at later times when compared with the MCF7 cultures (Figure 1B). These results are in complete agreement with previously published data that showed oxidative damage in H9c2 cells only after ≈24 h of DOXO exposure [50]. As shown in the immunoblotting images and the bar graph reporting the relative levels of band intensities, exposure to lactate caused a marked and statistically significant reduction in DNA damage.

This effect was remarkable at 16 h, when, in lactate-exposed cultures, the γ-H2AX band intensity became barely detectable; however, it also remained evident at 24 h, when the γ-H2AX band was reduced to about one-third compared with the control cultures. In agreement with these results, ROS evaluation in H9c2 cells treated with 1 μM DOXO for 16 h showed a marked reduction in lactate-exposed cells (Figure 6B); the extent of the effect (≈70% reduction) is in line with the results from the γ-H2AX immunoblotting evaluation in Figure 6A.

The same experiments were performed on differentiated H9c2 cultures (D-H9c2). The results are shown in Figure 7.

The immunoblotting evaluation in Figure 7A shows that, compared to their un-differentiated parental cultures, D-H9c2 appeared to be more prone to oxidative stress: the γ-H2AX band was more evident in untreated cultures, and its increase in response to DOXO treatment was already observed at 2 h. However, lactate exposure also successfully reduced DNA damage in D-H9c2 cells, and the extent of the protective effect measured at 16 and 24 h was found to be very similar to that observed in the parental culture (≈70% reduction in γ-H2AX band intensity). The higher level of oxidative stress in untreated D-H9c2 was confirmed with the experiments in Figure 7B, which showed ROS detection also in ≈10% of cells in the control cultures. Despite this increased vulnerability to oxidative damage, the coadministration of lactate successfully reduced DOXO-associated ROS (65% vs. 25% ROS-labeled cells in DOXO-exposed cultures, compared to cultures exposed to lactate/DOXO).

The increased vulnerability to oxidative stress of D-H9c2 could be explained by the metabolic switch (oxidative metabolism instead of glycolysis) that characterizes the differentiation process of normal cells and leads to reduced lactate production [51]. This hypothesis was confirmed with the experiment shown in Figure 8A. Compared to proliferating H9c2 cells, the differentiated culture showed a 40% reduction in lactate production.

Finally, for the cardiomyocyte cultures exposed to the combined lactate/DOXO treatment, we verified whether the observed oxidative damage protection (Figure 6 and Figure 7) could be ascribed to SOD1 and SOD2 proteins, as already demonstrated for MCF7 cells. The control and lactate-exposed H9c2 and D-H9c2 cells were treated with 1 μM DOXO for 16 h, the time-interval with the most evident reduction in oxidative damage in lactate-exposed cells (Figure 6 and Figure 7). The cells were then lysed, and the SOD1 and SOD2 levels were detected using immunoblotting. As shown in Figure 8B, in DOXO-treated H9c2 cells, both SOD1 and SOD2 proteins exhibited a 40–50% increased level when these cultures were previously exposed to 20 mM lactate. In D-H9c2 cells, the protective effect of lactate could be ascribed only to the SOD2 protein. This difference between the proliferating and differentiated culture can be explained by previous studies showing that in mature cardiomyocytes, the level of SOD1 is quite low and that, in these cells, the enzyme isoform primarily involved in oxidative damage protection is SOD2 [52]. From these experiments, we can conclude that the molecular mechanism underlying the oxidative damage protection induced by lactate is similar in both MCF7 cells and in cardiomyocyte cultures.

## 4. Discussion

DOXO and its analogs are widely used as a first-line chemotherapeutic treatment for different solid and hematologic malignancies. It was estimated that about 1 million cancer patients receive DOXO treatment every year [1,2,3]. The broad-spectrum activity of DOXO reflects the multiplicity of the identified intracellular targets for this drug. In addition to accounting for antineoplastic action, these targets are also involved in heavy DOXO side effects. Among these, cardiotoxicity proved to be the most relevant since it often leads to therapy exclusion or discontinuation for patients with compromised heart function [53]. A number of mechanisms have been hypothesized to contribute to the pathogenesis of DOXO-induced cardiotoxicity; among them, oxidative stress received by far the most attention since it can easily explain the mitochondrial damage and cardiomyocyte apoptosis frequently observed in experimental models used to characterize the molecular mechanisms underlying DOXO side effects [5,48,54]. Oxidative stress was found to be derived by metabolic processing of the DOXO molecule, which produces a semi-quinone derivative able to transfer its unpaired electrons to molecular oxygen, giving rise to ROS [55]. It is well known that ROS can cause membrane damage and mitochondrial dysfunction, leading to cell apoptosis. Recent studies also identified some molecular targets involved in DOXO-induced oxidative damage and confirmed the beneficial effects of antioxidant compounds in alleviating cardiotoxicity [56,57,58].

One of the most recently characterized mechanisms underlying DOXO action is its capacity to damage the chromatin structure and cause histone eviction, which results in transcriptome alterations [6,9]. This observation prompted studies aimed at evaluating the effect of the co-administration of DOXO with other compounds affecting gene expression, such as HDAC inhibitors.

Formerly regarded as a waste product, lactate is now considered one of the most evolutionarily ancient signaling molecules involved in the regulation of gene expression [59]. Preliminary observations suggested that this metabolite is capable of inhibiting HDAC. In 2019, Zhang et al. evidenced a different lactate-based histone acylation process: lactylation [13]. Histone lactylation was found to be directly linked to LDH activity, and, in recent years, it was shown to be induced in cell lines following exposure to agents or conditions increasing cellular lactate levels. This finding is in agreement with the original idea of Zhang et al., which proposed a “lactate clock” activated in cells by increased lactate levels and leading to histone lactylation [13]. Through this mechanism, lactate can modulate gene expression and coordinate metabolism with other physiological functions.

Neoplastic diseases are the most clinically relevant conditions characterized by activated glycolytic metabolism and increased lactate production; therefore, increased histone acetylation and/or lactylation can be reasonably hypothesized to contribute to the growth and invasiveness of cancer cells [60]. In line with this idea, in recent years, our research group tried to elucidate the role of lactate in affecting the response of cancer cells to antineoplastic treatment. During these studies, we found that increased lactate exposure can facilitate the onset of tamoxifen resistance in ER-positive breast cancer cells [15] and affect the efficacy of cisplatin [14]. The protective effects exerted by lactate on DNA integrity are coherent with the role of glycolytic metabolism in embryonic tissues [61]. For the treatment of cancer diseases, these effects could reduce the efficacy of chemotherapy. However, the results obtained from cultured cardiomyocytes suggest that when the antineoplastic mechanisms of the drug are not affected, the gene regulatory properties of lactate could be exploited to protect normal tissues from chemotherapy-associated damage.

## 5. Conclusions

In the present paper, we explored the effect of lactate on DOXO antineoplastic mechanisms and found that when MCF7 cells are exposed to DOXO in the presence of increased lactate levels, TOP2A poisoning (the main antineoplastic mechanism of the drug) is not affected. Interestingly, lactate-exposed MCF7 cells showed upregulated levels of SOD1 and SOD2 proteins and appeared to be protected from drug-induced oxidative damage. Oxidative damage plays a major role in cardiac toxicity caused by a drug. When the effect of lactate was evaluated on both proliferating and differentiated cardiomyocytes exposed to DOXO (Figure 6 and Figure 7), this metabolite succeeded in significantly reducing ROS generation and the DNA damage signatures caused by the drug.

Taken together, the results of our experiments suggest that, by relieving the oxidative damage in cardiomyocytes without significantly affecting the antineoplastic effect on cancer cells, short-term exposure to lactate prior to DOXO treatment could be considered as a possible attempt to increase the chemotherapeutic index of the drug. Our data encourage a better characterization of the potential effects of lactate in modulating the response of cancer cells to chemotherapy.

## Data Availability

All data are contained within this article.

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
