# Peer review of "Lactate Can Modulate the Antineoplastic Effects of Doxorubicin and Relieve the Drug’s Oxidative Damage on Cardiomyocytes"

_cancers, 2023, doi:10.3390/cancers15143728_

Round 1

Reviewer 1 Report

Doxorubicin (DOXO) has been widely used to treat various types of cancers, however the significant cardiotoxicity restricts its clinical use. This manuscript tried to explore whether administration of exogenous lactate could modulate the therapeutic response of DOXO and relieve its associated cardiotoxicity. The results showed that the combination of lactate could rescue DOXO-mediated oxidative damage to cardiomyocytes through activating the transcription of superoxide dismutase SOD1/2. At a cost, the combination impaired the anticancer activity of DOXO although not interfering DOXO-induced TOP2 poisoning, a major mechanism of action.

Comments:

1.        Fig. 1 only provided the information that lactate was produced differently among different cancer cells, which should not stand alone but be integrated into Fig. 2.

2.        The results shown in several figures were not detailly described and properly interpretated, which should be improved. For example, in Fig. 2 we noted that DOXO stimulated distinct DNA damage responses in different cancer cells. In MCF-7, DOXO induced an early DNA damage, which occurred between 2 h and 6 h. Interestingly, the expression of gamma-H2AX enhanced by DOXO in MCF-7 was then resumed to basal level, which was presumably attributed to the progressive LDHA induction and lactate synthesis. In lactate-exposed MCF7 cells, DOXO induced a lagged but sustained DNA damage response, which occurred after 6 h of stimulus. More significant lag of response was observed in those high-lactate expressing cells like MDA-MD-231 and CaCo2, both of which showed obvious DNA damages at 24 h after DOXO treatment. Thus, these findings suggest that lactate levels may determine DOXO-mediated DNA damage responses. The study should simultaneously analyze the dynamics of lactate upon DOXO treatment.

3.        It is interesting that the proliferating H9c2 and differentiated D-H9c2 displayed different capacities to respond to DOXO-induced DNA damage and oxidative stress. LDHA, lactate and SOD1/2 should also be documented in these cultures. Silencing SOD1/2 should be performed to determine whether they are critical for lactate-mediated protection of cardiomyocytes. Such distinct responses should be carefully interpretated and discussed.

4.        Long exposure of lactate could impair the anticancer activity of DOXO as shown in Fig. 6D. Thus, the conclusion should be improved.

5.        The statistical analysis for gamma-H2AX in Fig. 2B should be corrected.

6.        The immunofluorescence images appeared obscure, which should be replaced with high resolution figures. The scale bar should be marked in the immunofluorescence images.

The authors should carefully improve the description and interpretation on the acquired data 

Author Response

We thank the Reviewer for the time spent in evaluating our manuscript and for her/his helpful suggestions. We hope to have improved the description of our experiments.

  1. Figure 1 shows the difference in lactate production among the used cell lines and introduces the following experiments. According to the Reviewer’s request, we have now integrated it into Fig. 2 (Now, Fig. 1, panel A)
  2. The aim of the experiments shown in Fig. 2 (now Fig. 1B,C) was exactly to show that DOXO causes different DNA damage responses in cells characterized by different lactate production. As hypothesized and also demonstrated by the reported experiments, in cancer cells the early DNA damage signatures (2-6 h) can be ascribed to the DOXO induced oxidative stress. MDA-MB-231 and CaCo2 cells (two highly glycolyzing cultures) did not show early DNA damage and this finding is in line with the notion that highly glycolyzing cancers also show resistance to oxidative stress (see ref. 29 of the revised manuscript (RM)).

Oxidative stress was shown to be caused by the metabolic processing of DOXO molecule after cell entry (RM, lines 646-647, ref 55); this mechanism of action is compatible with transient DNA damage signatures. Our data (Fig. 4 of the RM) suggest that the epigenetic effects of lactate can be involved in the protection against this DOXO-induced effect. After this early and transient phase of DNA damage, other DOXO-induced effects take place at later times. The finding of evident DNA damage in lactate-exposed cells at later times (24 h) clearly demonstrate that lactate cannot offer protection against the later phase of DOXO damage, which is probably the most relevant one, as also suggested by the viability experiment of Fig. 6 (now Fig. 5).

In our opinion, the proposed study of lactate dynamics upon DOXO treatment would be biased by the DOXO antiproliferative effect which progressively arises during the incubation time and also inhibits metabolic reactions. Concerning the observed increase of LDHA in MCF7 cells, we have now briefly discussed this point (lines 290-295 of RM). On the basis of all our results, we can assume that, for its capacity of affecting gene expression, only the metabolite levels preceding treatment have the potential of modifying some aspects of cell response. We hope to have improved the description of these and of other results in the revised manuscript.

  1. Following the Reviewer’s requests, we tried to evidence LDHA in DOXO-exposed H9c2 cells. We used two different anti-LDHA antibodies (Cell Signaling #2012, the same used in the experiments of Fig. 1B,C; Epitomics #2468-1), which according to the manufacturer’s instructions also react with rat LDHA. No band was evidenced in both proliferating and differentiated H9c2 cultures. We can hypothesize that these cells only express LDHB, the heart LDH isoform. In the RM, we added new experiments on H9c2 cells (Figure 8 of RM). Fig.8A shows that proliferating H9c2 cells are characterized by a 40%-increased lactate production, compared to the differentiated cultures. This finding can explain the lower oxidative stress signatures observed in proliferating cells. Fig. 8B shows that also in cardiomyocytes the protective effect exerted by lactate on the DOXO-induced oxidative damage can be explained by increased levels of SOD proteins, even though we observed a difference between proliferating and differentiated H9c2 cultures (lines 600-631).
  2. In the RM the “Conclusions” paragraph has been changed, also according to the requests of the others Reviewers. As specified in the manuscript, long exposures to lactate ultimately lead to stable gene expression changes, causing chemoresistance (ref. 15). Chemoresistant cancer cells typically show upregulated glycolysis and, as a consequence, they are inherently exposed to high lactate levels; for this reason, it could be expected that also in these cells, additional, short-term exposures to lactate should not relevantly affect therapy outcome.
  3. As explained in the text and in the figure legend, the statistical analysis of Fig. 2B (now Fig. 1C) concerned the comparison between MCF7 cells and MCF7 cells exposed to lactate. Further comparisons with the other cell cultures were not considered since MDA-MB-231 and CaCo2 cells differ from MCF7 cells not only in the produced lactate levels, but in several biologically relevant properties. The aim of our study was to evidence statistically significant differences referable to lactate.
  4. We are aware that some of the immunofluorescence images appear not very clear; basically, this is true for lactate-exposed cells. However, this is not due to the low image resolution (300 dpi, as requested by the journal), but to the low fluorescence of lactate-exposed cell samples. We explained this problem in the legend of Fig.1 and added the scale bar in all the immunofluorescence images.
  5. We hope to have improved the description of our results in the revised manuscript.

Reviewer 2 Report

The manuscript entitled “Lactate can modulate the antineoplastic effects of doxorubicin and relieve the drug oxidative damage on cardiomyocytes”, written by Valentina Rossi , Marzia Govoni and Giuseppina Di Stefano, is of interest for the Journal’s Readers. It is also well written and the experiments appear to be of high quality. The references are well-chosen as well.

The manuscript may be accepted for publication after minor revision.

Row 59: “. Conclusions: besides contributing to elucidate the effects of the combined action of DOXO and lactate, our results propose a possible attempt to reduce the heavy drug cardiotoxicity, a major side effect leading to therapy discontinuation.” There is only one conclusion. Is there any format that can be respected?

Row 100: “c) to assess the level of acetylated Histone 3 in lactate-exposed MCF7 cells.Histone 3 or histone 3? Which is correct?

Row 122: „in the Figures, by ” ... which figures?

Row 127: „Bands’ intensities were evaluated with the aid of the ImageJ software.” Or better „The intensity of the bands was evaluated using the ImageJ software method.”? I do not know....

Row 130 : „Oxidative stress was evaluated by applying the 2’,7’-dichlorofluorescin diacetate (DCF-DA) assay.” Reference?

Row 354: As already mentioned in the Introduction, both lactate and DOXO have the potential of modifying gene expression.” Preferable “Both lactate and DOXO have the potential of modifying gene expression [Reference]”.

In “Conclusions”, please refer mostly to the co-administration of lactate and DOXO, which means your novel results.

Row 598: Is this „The protective effects exerted by lactate on DNA integrity are coherent with the role of glycolytic metabolism in embryonic tissues [55].” a conclusion of your research? It cn be moved to Discussion. Please, reformulate.

Author Response

We thank vary much the Reviewer for her/his appreciation of our work. We introduced the requested changes in the revised manuscript (see highlighted text) and re-wrote the “Conclusions” paragraph.

Reviewer 3 Report

In this study, the authors demonstrated whether the enhanced lactate levels which characterize neoplastic tissues can modify the response of cancer cells to DOXO. THe article sounds fine, which, however, requires some revisions, as stated below.

Suggest adding a schematic exploring the outline for better understanding of the content to the reader.

I suggest describing the outline at the end of the introduction explicitly.

Conclusions section looks simple. I suggest adding limitations in the conclusions section of the study.

Punctuation marks are irregularly used, suggest removing colons and semicolons redundant in some instances.

Better cite some recent refs in the past three years.

Instead of just citation, i suggest adding full details of the assay. for instance,  section 2.2

Minor.

Check the typological errors. By the way, please define all abbreviations when they appeared for the first time.

Add the city and country names of the companies in the experimental section.

Author Response

We thank vary much the Reviewer for her/his work in evaluating our manuscript. We introduced the requested changes in the revised manuscript (see highlighted text). Specifically:

  1. We added a scheme after the “Introduction” paragraph showing the outline of the described experiments.
  2. We re-wrote the “Conclusions” paragraph.
  3. We detailed the requested experimental procedure (paragraph 2.2).
  4. We checked the typological errors, defined the abbreviations at their first use and added the city and country names of the companies.
  5. Concerning references of the last three years, we have now cited 3 papers published in 2023 and discussing the problem of doxorubicin cardiotoxicity (n. 56-58). At present, 20 out of 61 references refer to papers published from 2020 onward.

Round 2

Reviewer 1 Report

Some errors should be corrected like:

line 15-16: "plays and important" should be "plays an important"

line 17: "A Background" should be "Background"

Some errors should be corrected like:

line 15-16: "plays and important" should be "plays an important"

line 17: "A Background" should be "Background"